# M-Mode Echocardiographic Measurements of Interventricular Septum, Left Ventricular Internal Diameter, and Left Ventricular Free-Wall Thickness in Normal Horses—A Meta-Analytical Study

**DOI:** 10.3390/ani13050809

**Published:** 2023-02-23

**Authors:** Mohamed Marzok, Mahmoud Kandeel, Hussein Babiker, Khalid M. Alkhodair, Alshimaa Farag, Hussam Ibrahim, Maged El-Ashker, Yousef Alghuwainem, Sabry El-khodery

**Affiliations:** 1Department of Clinical Sciences, College of Veterinary Medicine, King Faisal University, Al-Ahsa 31982, Saudi Arabia; 2Department of Surgery, Faculty of Veterinary Medicine, Kafrelsheikh University, Kafrelsheikh 33511, Egypt; 3Department of Biomedical Sciences, College of Veterinary Medicine, King Faisal University, Al-Ahsa 31982, Saudi Arabia; 4Department of Pharmacology, Faculty of Veterinary Medicine, Kafrelsheikh University, Kafrelsheikh 33511, Egypt; 5Department of Anatomy, College of Veterinary Medicine, King Faisal University, Al-Ahsa 31982, Saudi Arabia; 6Department of Internal Medicine, Infectious Diseases and Fish Diseases, Faculty of Veterinary Medicine, Mansoura University, Manosura 35516, Egypt

**Keywords:** cardiac diameters, horses, systematic review

## Abstract

**Simple Summary:**

The current meta-analysis offers a general perspective on the echocardiographic measurements of heart dimensions in sound Thoroughbred and Standardbred horses. The study was performed according to the guidelines of PRISMA. The echocardiographic findings varied from one study to the next due to the fact that there were multiple investigations. The meta-analysis indicates variations in results among different studies. This result should be considered when evaluating a horse for heart disease and each case should be evaluated independently.

**Abstract:**

The purpose of this study was to provide a systematic meta-analysis on echocardiographic measurements in normal Thoroughbred and Standardbred horses. The current systematic meta-analysis followed the Preferred Reporting Items for Systematic Reviews and Meta-Analyses (PRISMA). All the available published papers on the reference values of echocardiographic assessment via M-mode echocardiography were searched, and fifteen studies were finally selected for analysis. In both fixed and random effect, the confidence interval (CI) for the interventricular septum (IVS) was 2.8–3.1 and 4.7–7.5; for the left ventricular free-wall (LVFW) thickness, it was 2.9–3.2 and 4.2–6.7; and for the left ventricular internal diameter (LVID), it was −5.0–4.6 and −10.0–−6.7, respectively. For IVS, the Q statistic, I-squared, and tau-squared were 925.3, 98.1, and 7.9, respectively. Similarly, for LVFW, all the effects were on the positive side of zero, with a range of 1.3–68.1. The CI indicated a significant variation among the studies (fixed, 2.9–3.2; random, 4.2–6.7). The z-values of LVFW for fixed and random effects were, respectively, 41.1 (*p* < 0.001) and 8.5 (*p* < 0.001). However, the Q statistic was 886.6 (*p* < 0.001). Moreover, the I-squared was 98.08, and the tau-squared was 6.6. By contrast, the effects of LVID fell on the negative side of zero, (2.8–83.9). The present meta-analysis provides an overview of the echocardiographic measurements of cardiac diameters in healthy Thoroughbred and Standardbred horses. The meta-analysis indicates variations in results among different studies. This result should be considered when evaluating a horse for heart disease and each case should be evaluated independently.

## 1. Introduction

Echocardiography is a non-invasive procedure that permits the thorough visualization of cardiac chambers as well as their valves and is helpful in the diagnosis and monitoring of animals with suspected or known heart diseases [1]. Echocardiography provides an evaluation of cardiac output, ejection fraction, diastolic function, and fractional shortening [2]. Moreover, it helps to diagnose cardiac diseases, including atrial septal defects, atrioventricular valve stenosis, aortocardiac fistula and hypertrophic cardiomyopathy, and dilated cardiomyopathy [1,3,4].

M-mode echocardiography is used to obtain high-resolution real-time images of cardiac structures [5,6]. It was the first applicable procedure in horses in the late 1970s [7]. Via M-mode echocardiography, the heart chambers, myocardium, valves, pericardium, and great vessels can be easily visualized [8]. M-mode echocardiography has a high sampling rate when compared with 2D-mode and is superior to real-time images in recording subtle changes in the wall and valve motion and is used for the assessment of the size and function of the left ventricle [5]. In-depth reports on M-mode echocardiography in foals, adult horses, and ponies, including measures of heart size, have been provided [9,10,11,12,13,14,15].

Echocardiography’s ability to measure cardiac diameters is widely regarded as a pivotal technique for assessing the severity and prognosis of cardiac disease and the heart’s reaction to physical exertion [16]. Indeed, in horses, the effects of training [17,18], growth [12,19], body weight [19,20,21,22], animal’s gender [11,16,18,23], and animal breed [20,24] on echocardiographic measurements have been described. However, these effects have never been statistically tested. The majority of investigations to examine the intra- and inter-observer repeatability of equine echocardiographic measures were conducted in horses and donkeys [2,25].

Meta-analysis is defined as a quantitative, epidemiological study design used to systematically assess previous research studies to reach conclusions about that body of research [26]. Meta-analyses have attracted much attention from the general public and have become increasingly popular in the biomedical field [27]. The medical and statistical literature contains numerous references to the theory and correct methodology for meta-analysis [28].In addition to obtaining definitive conclusions regarding the research topic at hand, meta-analysis estimates the real effect of a treatment or exposure on a certain outcome by combining data from many trials [27,29]. Subsequently, a consolidated and quantitative review of such large and complex studies is obtained [26]. The examination of variability or heterogeneity in the results obtained by previous studies is also a critical outcome of meta-analyses [30].

Meta-analysis assists to overcome the lack of the statistical capacity to reach concrete conclusions in independent studies, as well as the failure to appropriately analyze the differences in the risk of extremely uncommon adverse events in large-scale studies [31]. When the results of several studies, even if they are incompatible with one another, are systematically integrated, the real magnitude of the effect may be more precisely defined [32].

In fact, extensive meta-analysis studies on echocardiographic measurements in humans have been carried out [33]. However, in the veterinary field, limited studies were carried out on canine echocardiography [34]. According to the authors’ knowledge, there are no available studies in the literature on the meta-analysis of echocardiographic parameters in horses. Consequently, the objective of this systematic review was to present a meta-analysis on the measurements of the interventricular septum (IVS), the left ventricular free-wall (LVFW) thickness, and the left ventricular internal diameter (LVID), in healthy Standardbred and Thoroughbred horses.

## 2. Materials and Methods

### 2.1. Selected Studies

The reference values for the echocardiographic evaluation of heart diameters in Thoroughbred and Standardbred horses were generated in this meta-analysis.

### 2.2. Types of Reference Individuals

None of the horses investigated had any cardiac abnormalities or pathological conditions that might affect cardiac function or size. Any level of sample size was acceptable.

### 2.3. Inclusion and Exclusion Criteria

#### 2.3.1. Inclusion Criteria


-For papers written in a foreign language, the availability of the English version of the reports;-Measurements were obtained via M-mode echocardiography;-Measurements were obtained via the right parasternal short-axis view (RPSAX);-Measurements were obtained at end-systole and end-diastole.


#### 2.3.2. Exclusion Criteria


-Measurements taken from other different breeds;-Methods of examination other than RPSAXs;-Incomplete parameters of cardiac diameters;-Non-English published papers.


### 2.4. Search Strategy and Selection of Studies

The goal was to find all published publications dating back to the early days of regular echocardiography. The authors searched the PubMed, Ovid, Sage, BESCO, CAB, Scopus, and ISI web of knowledge databases from the beginning of time until July 2018 using the search phrases ECHOCARDIOGRAPHY (title/abstract) AND (“NORMAL VALUES”) (title/abstract) OR (“REFERENCE VALUES”) (title/abstract). This method was supplemented by citation evaluation, Google Scholar searches, expert suggestions, and hand-searching. Using EndNote, a reference application, we integrated the database results (version X7; Thomson Reuters). The articles included in this study are shown in Figure 1.

### 2.5. Data Extraction and Analysis

The following information was extracted using a data extraction form: the study’s year, sample size, and provided summary statistics (i.e., the standard difference in means, standard error, variance, Hedges’s g, and the difference in means), types of breed, and the measurements and methods of cardiac diameters during the cardiac cycle (Table 1).

### 2.6. Quality Assurance

The current systematic meta-analysis followed the Preferred Reporting Items for Systematic Reviews and Meta-Analyses (PRISMA) [40]. All the available published papers on the reference values of the echocardiographic assessment of cardiac diameters in normal healthy Thoroughbred and Standardbred horses were included, to minimize publication bias.

### 2.7. Statistical Analysis

For statistical analyses, a commercial software program for meta-analysis was used (comprehensive meta-analysis version 2, USA). Descriptive statistics were applied to present the mean values of echocardiographic parameters in the finally selected papers. The variables meta-analysis used were fixed and random-effect models, 95% confidence intervals, effect size, heterogeneity, and weight. The effect size was determined using a standardized Z statistic and *p*-value [41]. Heterogeneity was assessed using Cochrane’s Q test with a significant value of *p* < 0.05, and the I^2^ statistic was used to define the % of true heterogeneity among the analyses. The I^2^ statistic was used to estimate the degree of heterogeneity, which refers to the total variation depending on the Q statistic and the number of trials (K). In fact, a negative value of I^2^ was considered equal to zero, and consequently, the I^2^ statistic ranged between 0% and 100%, and a value equal to or more than 50% was considered heterogeneous [42]. With values of 25%, 50%, and 75%, low, moderate, and high degrees of heterogeneity were identified, respectively [30]. Study weight was calculated as the base inverse square of the standard error of the effect of each trial. Forest plots were used to present the means and their confidence intervals in a graphic manner, and heterogeneous degrees were explored. Meta-regression was performed to study the effect of age, gender, and body weight on the echocardiographic measurements, and a result with a *p*-value < 0.05 was considered significant.

## 3. Results

The final model of meta-analysis for the measurement of IVS, LVID, and LVFW in healthy Standardbred and Thoroughbred horses is presented in Table 2 and Figure 2, Figure 3 and Figure 4.

The results revealed that all the effects of IVS were on the positive side of zero, in the range of 1.36–66.08. The 95% confidence interval (CI) supported this finding in both fixed and random effects, indicating significant variations among the studies (fixed, CI 95%: 2.8–3.1; random, CI 95%: 4.7–7.5 (Table 2, Figure 2). For the relative weight, the study by Al-Haidar et al. (2013) was assigned a relative weight of 0.01%, while that of Leadon et al. (1991) was assigned a relative weight of 57.97%. In our study, the test of the null hypothesis (two-tailed) showed that the z-values of IVS for fixed and random effects were 39.5 (*p* < 0.001) and 8.7 (*p* < 0.001), respectively. For heterogenicity, the Q statistic was 925.3, compared with the expected value of 17 (*p* < 0.001); the I-squared was 98.1, and the tau-squared was 7.9.

On the other side, all the effects of LVID fell on the negative side of zero, in the range of 2.8–83.9. The 95% CI indicated a significant variation among the studies (fixed, 95% CI 5.08 to 4.6; random, 95% CI 10.08–6.7) (Table 2, Figure 3). Under a fixed effect, the point estimate was −4.8, with a standard error of 0.102. By contrast, under a random effect, the point estimate was −8.3, with a standard error of 0.8. For the relative weight, the study by Al-Haidar et al. (2013) was assigned a relative weight of 0.01%, while that of Leadon et al. (1991) was assigned a relative weight of 51.57%. For the test of the null hypothesis (two-tailed), the z-values for the fixed and random effects were 48.09 (*p* < 0.001) and 9.7 (*p* < 0.001), respectively. For heterogenicity, the Q statistic was 844.949 (*p* < 0.001), and the I-squared and tau-squared values were 97.988 and 11.977, respectively.

Similarly, for LVFW, all the effects were on the positive side of zero, in the range of 1.350 to 68.176. The 95% CI indicated a significant variation among studies (Table 2, Figure 4). Under a fixed effect, the point estimate was 3.1, with a standard error of 0.07 and a 95% CI of 2.93.2. However, under a random effect, the point estimate was 5.4, with a standard error of 0.6 and a 95% CI of 4.2 to 6.7. For the test of the null hypothesis (two-tailed), the z-values for the fixed and random effects were 41.114 (*p* < 0.001) and 8.545 (*p* < 0.001), respectively. However, for heterogenicity, the Q statistic of LVFW was 886.648, compared with the expected value of 17 (*p* < 0.001), while the I-squared was 98.083, and the tau-squared was 6.680.

## 4. Discussion

In the present study, we conducted a systematic meta-analysis on the echocardiographic measurements of IVS, LVID, and LVFW, in Thoroughbred and Standardbred horses. For IVS, LVID, and LVFW, considering both types of effects (fixed and random), there was a variation among the results of the measurements. It is speculated that the controversy about the reliability of studies is due to the precision of the technique used. This finding is consistent with that previously reported [43]. In the present results, there was a non-significant effect of body weight, age, breed, and gender on the measurements. The combined effect of such factors with other unknown variables may be the cause of variations in measurements. It has also been reported that training and growth can affect echocardiographic measurements [17,18].

For IVS, the confidence interval (CI) of 95% in the study of Al-Haidar et al. [2] was about six times as wide as that in another study by Al-Haidar et al. [35], and the values in eleven other studies fell somewhere in between. Al-Haidar et al. [2] showed that the effect of size was reduced to 2.9 in the fixed-effect model, in which the weights dominated in the current study. The overall effect estimate was accurate, with a tiny within-study error for IVS of 0.07. Currently, just 15 research were analyzed, and their impact sizes varied widely. Our standard error of IVS (0.6) was almost six times the fixed-effect value, indicating that our estimate of the mean impact was not exact. It has been reported that a within-study error is an accurate indicator to assess the mean impact [44].

For LVID, the 95% CI reported by Pipers and Hamlin [7] was about three times as wide as the one recorded by Brown et al. [22], while the values in fifteen other studies fell somewhere in between. Interestingly, the study of Brown et al. [22] had a low effect size. Under a fixed effect, the effect size decreased to 4.8%. However, the 95% CI was 8.3 when random factors were considered. The LVID research had a slight within-study error due to a large number of individuals (15 in each group) (0.1). The standard error for the mean effect was 0.864, which was about three times the standard error for the fixed-effect value, indicating that the mean impact was not particularly precise.

For LVFW, the 95% CI stated by Slater and Herrtage [20] was about three times as wide as the one stated by Al-Haidar et al. [35]. Under the fixed-effect model, the effect size was reduced to 3.144, but it was 5.498 under the random-effect model. The within-study error for LVFW was small (0.07), indicating an accurate estimate of the combined effect. The mean effect sizes varied, where the standard error was 0.643, which was about three times that of the fixed-effect value.

The confidence interval (CI) of 95% for IVS supported these findings in both fixed and random effects, indicating a significant variation among the studies (fixed, 95% CI 2.8–3.1; random, 95% CI 4.7–7.5) and for LVID (fixed, 5.0 to 4.6; random, −10.0–−6.7) and for LVFW (fixed, 2.9–3.2; random, 4.2–6.7). The measurements of the three variables varied among the studies, which may be due to the wide range of horse breeds and ages in most of the studies.

The relative weight is the average of weights as a proportion of total weights, with all relative weights added up to 100% [45]. For IVS, the study by Leadon et al. (22) was given 58% of the weight under the fixed-effect model but only 6% of the weight under the random-effect model despite having a high sample size (n = 600 for each group). Under the fixed-effect paradigm, we assumed that all the studies and the existing study had the same value. A more accurate estimate was found in the study by Leadon et al. [23]. On the other hand, it was assumed that each study estimated a different impact in the random-effect model. The same study provided an accurate estimate of the IVS population, but because it was just one among several, we did not want it to dominate the analysis. As a result, we gave it 6% of the relative weight, which was more than the weight we gave it under fixed effects but lower than previous studies.

For LVID, the study of Leadon et al. (1991) [23] was assigned 52% of the weight under the fixed-effect model but only 6% of the weight under the random-effect model. The above-mentioned study provided an accurate estimate of its population. In our study, we assigned it 6% of the weight. This was more than the values in the other studies but not as dominant a weight as that we gave it under fixed effects. Similarly, for LVFW, the study by Leadon et al. [23] was assigned 46% of the weight under the fixed-effect model but only 6% of the weight under the random-effect model.

Regarding heterogeneity, under both fixed (common) and random (true) effects, the null hypothesis was that the effect would be zero. The null hypothesis was tested using the z-value, which was computed as Hedges’s g/standard error (G/SE) for the corresponding model [46]. In our study, the z-values for the fixed and random effects of IVS were 39.519 (*p* < 0.001) and 8.793 (*p* < 0.001), and for LVID, they were 48.090 (*p* < 0.001) and 9.719 (*p* < 0.001). For LVFW, the z-values were 41.114 (*p* < 0.001) and 8.545 (*p* < 0.001), respectively.

In the present study, the Q statistic for IVS was 925.307, while for LVID, it was 844.949, and for LVFW, it was 886.648, compared with the expected value of 17 (*p* < 0.001). The Q statistic implied the observed dispersion, whereas the null hypothesis for heterogeneity argued that the studies assigned a common effect size. Therefore, the degrees of freedom were assumed to be equal to the Q statistic [47].

While the Q statistic is used to test the null hypothesis that there is no dispersion across effect sizes, the I-squared and tau-squared are valuable parameters to evaluate [48]. The I-squared of IVS was 98.163, while that of LVID was 97.988, and that of LVFW was 98.083, indicating that the real variations in effect sizes accounted for 90% of the apparent variance among the studies. On the basis of random error, only 10% of the observed variation may be predicted. However, the tau-squared of IVS was 7.905; that of LVID was 11.977; and that of LVFW was 6.680. This is the variance “between studies” that was used in computing the weights.

Because they are derived using fixed-effect weights, the Q statistic and tau-squared are normally given in fixed-effect models but not in random-effect models. Q statistic solves the question of whether the fixed-effect model fits the data in the fixed-effect study (i.e., whether it is sufficient to suppose that the tau-squared is really zero). To give weights, however, the tau-squared is really set to zero [30]. Unfortunately, the Q test is only employed in meta-analysts to inform on the presence or absence of heterogeneity; it does not reveal the degree of such heterogeneity. In a meta-analysis investigation, the I-squared index has been proposed to measure the degree of heterogeneity [49]. In our study, the statistical differences between the studies may be attributed to the sample size, body weight, breed, and gender.

The limitations of the present investigation should be acknowledged. First, like other meta-analyses reporting reference values, remarkable heterogeneity was an innate limitation, although we attempted to explore the source of heterogeneity and define reference values with detailed information. Second, not all variables were comparable, as only IVS, LVID, and LVFW were presented via M-mode echocardiography. These variables were selected for investigation in all the studies. However, in other meta-analytical studies in humans and animals, a wide range of cardiac diameters was included [33,50,51]. Moreover, to minimize the limitations in our study, the checklist of PRISMA, which is considered a standard technique, was used.

## 5. Conclusions

This is the first meta-analysis that summarizes the results of the currently available studies looking at the identification of normative values for IVS, LVID, and LVFW, as assessed via M-mode echocardiography in horses. All the variables of this meta-analysis indicated variations in the results among the different studies. This result should be considered when evaluating a horse for heart disease, and each case should be evaluated independently.

## Figures and Tables

**Figure 1 animals-13-00809-f001:**
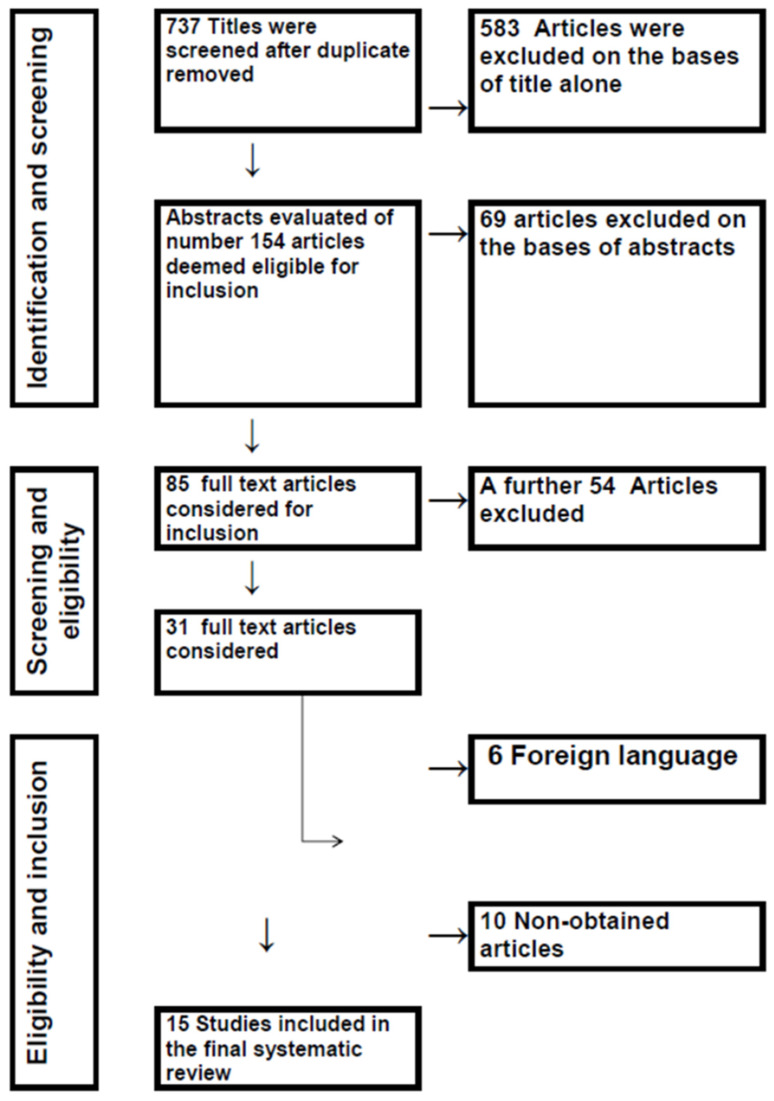
Results of the literature search and disposition of echocardiographic articles screened for inclusion.

**Figure 2 animals-13-00809-f002:**
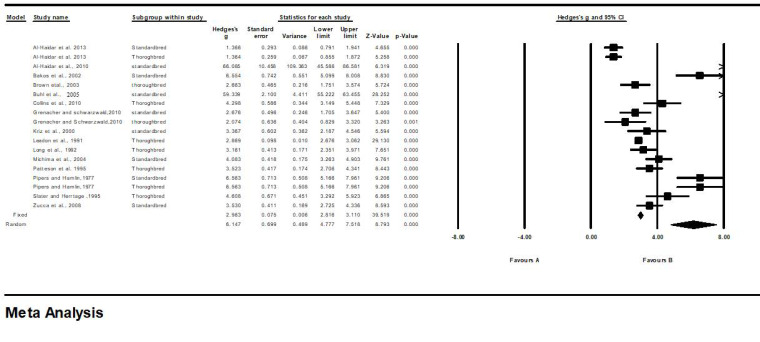
Forest plot for interventricular septum (IVS) measurements in Thoroughbred and Standardbred horses. Studies and their respective citations: Al-Haidar et al., 2013 [35]; Al-Haidar et al., 2010 [2]; Bakos et al., 2002 [21]; Brown et al., 2003 [22]; Buhl et al., 2005 [18]; Collins et al., 2010 [36]; Grenacher and Schwarzwald, 2010 [37]; Kriz et al., 2000 [38]; Leadon et al., 1991 [23]; Long et al., 1992 [39]; Michima et al., 2004 [8]; Patteson et al.,1995 [6]; Pipers and Hamlin, 1977 [7]; Slater and Herrtage, 1995 [20]; Zucca et al., 2008 [16].

**Figure 3 animals-13-00809-f003:**
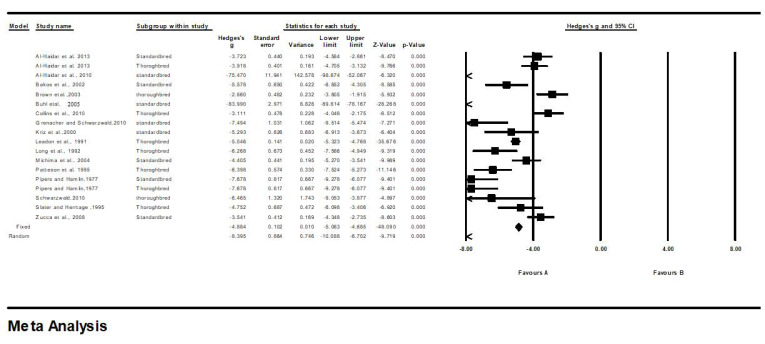
Forest plot for left ventricular internal diameter (LVID) measurements in Thoroughbred and Standardbred horses. Studies and their respective citations: Al-Haidar et al., 2013 [35]; Al-Haidar et al., 2010 [2]; Bakos et al., 2002 [21]; Brown et al., 2003 [22]; Buhl et al., 2005 [18]; Collins et al., 2010 [36]; Grenacher and Schwarzwald, 2010 [37]; Kriz et al., 2000 [38]; Leadon et al., 1991 [23]; Long et al., 1992 [39]; Michima et al., 2004 [8]; Patteson et al.,1995 [6]; Pipers and Hamlin, 1977 [7]; Slater and Herrtage, 1995 [20]; Zucca et al., 2008 [16].

**Figure 4 animals-13-00809-f004:**
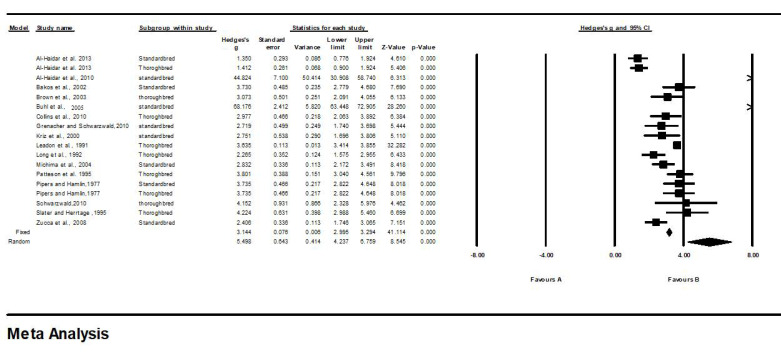
Forest plot for left ventricular free-wall (LVFW) thickness measurements in Thoroughbred and Standardbred horses. Studies and their respective citations: Al-Haidar et al., 2013 [35]; Al-Haidar et al., 2010 [2]; Bakos et al., 2002 [21]; Brown et al., 2003 [22]; Buhl et al., 2005 [18]; Collins et al., 2010 [36]; Grenacher and Schwarzwald, 2010 [37]; Kriz et al., 2000 [38]; Leadon et al., 1991 [23]; Long et al., 1992 [39]; Michima et al., 2004 [8]; Patteson et al.,1995 [6]; Pipers and Hamlin, 1977 [7]; Slater and Herrtage, 1995 [20]; Zucca et al., 2008 [16].

**Table 1 animals-13-00809-t001:** Descriptive statistics for M-mode echocardiographic measurements of cardiac diameters in Thoroughbred and Standardbred horses.

Study Name		Sample Size	Breed	Age Range (Year)	Gender	Body Weight (Kg)	Variables					
							IVS		LVID		LVFW	
							Systole	Diastole	Systole	Diastole	Systole	Diastole
1	Al-Haidar et al., 2013 [35]	36	Thoroughbred	2–21 Y (5.24 ± 2.34)	Mares Geldings Stallions	86-6 (452.9 ± 52.3)	3.99 ± 1.03	2.57 ± 1.03	6.20 ± 1.04	10.28 ± 1.02	3.16 ± 1.03	2.14 ± 1.03
		28	Standardbred	1.7–21 Y (8.9 ± 5.24)	Mares Geldings Stallions	370–541 (472.45 ± 36.31)	3.99 ± 1.02	2.57 ± 1.03	6.11 ± 1.03	9.98 ± 1.02	3.55 ± 1.03	2.14 ± 1.03
2	Al-Haidar et al., 2010 [2]	10	Standardbred	1.5–25.6 Y (10.10 ± 7.6)	5 Mares 5 Geldings	120–662 kg (459.8 ± 128.8)	4.05 ± 0.02	2.67 ± 0.02	6.25 ± 0.05	10.19 ± 0.05	3.48 ± 0.04	2.00 ± 0.02
3	Bakos et al., 2002 [21]	23	Standardbred	2–16 Y (Mean 6)	8 Mares 7 Stallions 8 Geldings	350–490 (Mean 427)	4.7 ± 0.3	3.0 ± 0.2	7.0 ± 0.6	10.7 ± 0.7	3.9 ± 0.4	2.7 ± 0.2
4	Brown et al., 2003 [22]	17	Thoroughbred	10.39	Males	450.7	5.26 ± 0.63	3.66 ± 0.54	8.65 ± 1.61	13.92 ± 1.97	5.29 ± 0.89	3.1 ± 0.42
5	Buhl et al., 2005 [18]	200	Standardbred	3–8	Mares	477 to 540	3.50 ± 0.02	2.56 ± 0.01	7.66 ± 0.04	11.47 ± 0.05	3.47 ± 0.02	2.39 ± 0.01
6	Collins et al., 2010 [36]	19	Thoroughbred	Up to 4 months	11 Males 8 Females	190 (182.10 ± 15.16)	3.09 ± 0.29	1.96 ± 0.22	5.52 ± 0.67	7.78 ± 0.75	2.33 ± 0.32	1.57 ± 0.15
7	Grenacher and Schwarzwald, 2010 [37]	15	Standardbred	>2 Y	Geldings Females	539 ± 31	3.8 ± 0.55	2.5 ± 0.38	5.3 ± 0.89	10.2 ± 1.32	2.7 ± 0.43	1.7 ± 0.31
		7	Thoroughbred	>2 Y	Geldings Females	548 ± 59	4 ± 0.52	3.0 ± 0.37	6.6 ± 0.87	10.9 ± 1.36	3.9 ± 0.47	2.2 ± 0.27
8	Kriz et al., 2000 [38]	13	Standardbred	3–4 Y	Geldings	411 ± 10	4.32 ± 0.32	3.19 ± 0.33	8.07 ± 0.69	11.76 ± 0.66	3.73 ± 0.51	2.51 ± 0.33
9	Leadon et al., 1991 [23]	600	Thoroughbred	Mean 19.3 ± 1.2 months	442 Males 158 Females	437 ± 35	2.0 ± 0.5	2.9 ± 0.4	6.3 ± 0.9	10.5 ± 0.9	3.8 ± 0.6	2.4 ± 0.4
10	Long et al., 1992 [39]	26	Thoroughbred	2–17 Y	3 Stallions 5 Mares 18 Geldings	432–648 (Mean 517)	4.55 ± 0.55	3.02 ± 0.39	7.35 ± 0.72	11.90 ± 0.71	3.96 ± 0.93	2.39 ± 0.26
11	Michima et al., 2004 [8]	35	Standardbred	5–18 Y	Males Females	415.51 ± 36.76	4.17 ± 0.42	2.68 ± 0.29	5.94 ± 0.96	9.72 ± 0.72	4.23 ± 0.69	2.69 ± 0.32
12	Patteson et al., 1995 [6]	29	Thoroughbred	3–15 y (M 7.71 Y)	22 Females 16 Geldings	420–617 (M 517)	4.21 ± 046	2.85 ± 0.278	7.45 ± 0.615	11.92 ± 0.76	3.85 ± 0.414	2.32 ± 0.382
13	Pipers and Hamlin, 1977 [7]	25	Standardbred and Thoroughbred	Mean 3.8 Y	Males Females	300 kg	4.7 ± 0.3	3.0 ± 0.2	7.0 ± 0.6	10.7 ± 0.3	3.9 ± 0.4	2.7 ± 0.2
14	Slater and Herrtage, 1995 [20]	16	Thoroughbred	4–16 Y (mean 7 Y)	11 Mares 5 Geldings	450–620 (490)	4.6 ± 0.2	2.6 ± 0.2	7.3 ± 0.8	11.2 ± 0.8	3.8 ± 0.3	2.5 ± 0.3
15	Zucca et al., 2008 [16]	30	Standardbred	3–9 Y (mean 3.8 ± 1.6 Y)	11 Females 17 Male 2 Geldings	340–498 kg (435 ± 36 kg)	4.48 ± 0.36	3.10 ± 0.41	7.42 ± 1.05	11.64 ± 1.29	3.64 ± 0.52	2.55 ± 0.36

IVS = interventricular septum; LVID = left ventricular internal diameter; LVFW = left ventricular free wall; Y = year.

**Table 2 animals-13-00809-t002:** Final meta-analysis model for reference values of echocardiographic measurements in Thoroughbred and Standardbred horses.

Variables	Model	Effect Size and 95% Confidence Intervals						Test of Null (2-Tailed)		Heterogeneity				Tau-Squared
		Number Studies	Point Estimate	Standard Error	Variance	CI 95% Lower Upper		Z-Value	*p*-Value	Q-Value	Df (q)	*p*-Value	I-Squared	
IVS	Fixed	15	2.9	0.1	0.01	2.8	3.1	39.5	<0.001	925.3	17	<0.001	98.1	7.9
	Random	15	6.1	0.6	0.4	4.7	7.5	8.7	<0.001	-	-	-	-	-
LVID	Fixed	15	−4.8	0.1	0.01	−5.0	−4.6	−48.1	<0.001	844.9	17	<0.001	97.9	11.9
	Random	15	−8.3	0.8	0.7	−10.0	−6.7	−9.7	<0.001	-	-	-	-	-
LVFW	Fixed	15	3.1	0.1	0.01	2.9	3.2	41.1	<0.001	866.6	17	<0.001	98.1	6.6
	Random	15	5.4	0.6	0. 4	4.2	6.7	8.5	<0.001	-	-	-	-	-

IVS = interventricular septum; LVID = left ventricular internal diameter; LVFW = left ventricular free wall.

## Data Availability

Not applicable.

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
