# Peer review of "M-Mode Echocardiographic Measurements of Interventricular Septum, Left Ventricular Internal Diameter, and Left Ventricular Free-Wall Thickness in Normal Horses—A Meta-Analytical Study"

_animals, 2023, doi:10.3390/ani13050809_

Round 1

Reviewer 1 Report

General remark: With this title a practitioner will start to read the abstract but stop to read it further. This manuscript could give very interesting information for practitioners, however, at this moment, the article is missing ‘basic’ information. If you could reorganize the information in the manuscript – add more ‘basic’ information in the abstract and in the ‘Introduction part’ then you will have a broader type of public that will read, and be interested, in the article.

 Detailed comments:

Title:

It is only for measurements performed via M-mode via the right parasternal approach – the title is therefore misleading

Abstract

Line 17: (LVID) dimension or diameter?

Line 18: dimensions only diameters or also area (further in the article it is only diameters)

Line 20-21: check sentence

Line 26: change capital letter to small letter, add “space”

The introduction needs information why you do this meta-analysis. For example, information that is now in the discussion part should be in the introduction part.

Why did you choose the measurements performed via M-mode?

Line 46: coronary artery disease in horse?

Information/ explanations that are addressed in the discussion part should already be in the abstract

Author Response

Dear Professor,

Thank you very much for the effort made to review our manuscript and for the valuable comments. I would like to confirm that all of these valuable comments have been considered completely while revising the paper.

General remark: With this title a practitioner will start to read the abstract but stop to read it further. This manuscript could give very interesting information for practitioners, however, at this moment, the article is missing ‘basic’ information. If you could reorganize the information in the manuscript – add more ‘basic’ information in the abstract and in the ‘Introduction part’ then you will have a broader type of public that will read, and be interested, in the article.

 Detailed comments:

Comment

Title:

It is only for measurements performed via M-mode via the right parasternal approach – the title is therefore misleading

Response

Yes, the title is corrected as recommended.

Comment

Abstract

Line 17: (LVID) dimension or diameter?

Line 18: dimensions only diameters or also area (further in the article it is only diameters)

Response

Yes, corrections were made for each variable.

Comment

Line 20-21: check sentence

Response

Yes, the sentence was revised.

Comment

Line 26: change capital letter to small letter, add “space”

Response

Yes, correction was done.

 Comment

The introduction needs information why you do this meta-analysis. For example, information that is now in the discussion part should be in the introduction part.

Response

Yes, done.

Comment

Why did you choose the measurements performed via M-mode?

Response

As M-mode echocardiography is a more reliable method for evaluation of cardiac measurements. In addition, in the criteria of selection, M-mode was added.

Comment

Line 46: coronary artery disease in horse?

Response

Yes, this sentence was revised as directed.

Comment

Information/ explanations that are addressed in the discussion part should already be in the abstract

Response

Yes, discussion and abstract were revised as a whole

Reviewer 2 Report

Review animals-2057928

The paper covers a lot of interesting information, however, a lot of information should be clarified, also for the wider audience. The aim of the study should be underlined. The methods part needs details. The presentation of the result section is not clear enough. Some aspects of data selection and statistical analysis should be rewritten. The discussion part includes many results. The conclusions are not specific enough.

Detailed remarks:

L 35 – the selection of just these breeds should be written in the method part and divided –why just these horses? Are they comparable?

L 48 – the M-mode echocardiography is described in the introduction, but it is not mentioned in the method part that it was a selection key. See table 1.

L 58-59 – the effects are given here, but are not discuss how the authors deal with them. There is also no limitation part in the discussion.

L 65 – definite conclusion is written here,  definite conclusions were not written

L79-80 – that is rather part of the aim, instead of that here the M-mode should be describe

L 83 – “any level of sample size should” be motivated

L 105-109 – it should be clearly stated in the method part that such information was extracted, but not used, not evaluated as a factor.

L 115-122 – all tests that were used and their results presented should be described here in detail, not only these treated as the main output – you present the results of them.

L 118-120 – this is not clear –effect of what? Please be specific as possible.

Table 1 – should be moved after point 2.5 and cited there. The situation where the result part consists of four tables almost without the text is not acceptable for reading.

Figure 1 – should be moved to the material part.

Table 2 – should be described better in the result part. The p-value of the test of null and heterogeneity being 0.000 in all cases should be given in detail. These values should be discussed in the discussion part, as well as the fitting of the model could be discussed more especially as some literature on fixed/random effects exists.

Figures 2,3, and 4 – effects should be described better, they are not clear from the methodology and the figures as well.

L 197 – homogeneity of data should be discussed

L 204 –all effects? What effects?

L237-238- to method part as well

L 258 – and what is the conclusion from this part?

L 278 – these effects were not discussed. Limitation part of the discussion is lacking.

Conclusions are weak.

Author Response

Reviewer 2

Dear Professor,

Thank you very much for the effort made to review our manuscript and for the valuable comments. I would like to confirm that all of these valuable comments have been considered completely while revising the paper.

Review animals-2057928

The paper covers a lot of interesting information, however, a lot of information should be clarified, also for the wider audience. The aim of the study should be underlined. The methods part needs details. The presentation of the result section is not clear enough. Some aspects of data selection and statistical analysis should be rewritten. The discussion part includes many results. The conclusions are not specific enough.

Detailed remarks:

Comment

L 35 – the selection of just these breeds should be written in the method part and divided –why just these horses? Are they comparable?

Response

Yes, we added this information in methodology, because all studies of echocardiography were conducted on such breeds.

Comment

L 48 – the M-mode echocardiography is described in the introduction, but it is not mentioned in the method part that it was a selection key. See table 1.

Response

Yes, we added this information in the inclusion criteria

L 58-59 – the effects are given here, but are not discuss how the authors deal with them. There is also no limitation part in the discussion.

Comment

L 65 – definite conclusion is written here, definite conclusions were not written

Response

Yes, conclusion was revised as recommended

Comment

L79-80 – that is rather part of the aim, instead of that here the M-mode should be describe

Response

Yes, we added this information as the inclusion criteria.

L 83 – “any level of sample size should” be motivated

Comment

L 105-109 – it should be clearly stated in the method part that such information was extracted, but not used, not evaluated as a factor.

Response

Yes, these data are presented completely in table 1

Comment

L 115-122 – all tests that were used and their results presented should be described here in detail, not only these treated as the main output – you present the results of them.

Response

Yes, done

Comment

L 118-120 – this is not clear –effect of what? Please be specific as possible.

Response

Yes, the sentence was revised as recommended.

Comment

Table 1 – should be moved after point 2.5 and cited there. The situation where the result part consists of four tables almost without the text is not acceptable for reading.

Response

Yes, done.

Comment

Figure 1 – should be moved to the material part.

Response

Yes, done.

Table 2 – should be described better in the result part. The p-value of the test of null and heterogeneity being 0.000 in all cases should be given in detail. These values should be discussed in the discussion part, as well as the fitting of the model could be discussed more especially as some literature on fixed/random effects exists.

Comment

Figures 2,3, and 4 – effects should be described better, they are not clear from the methodology and the figures as well.

Response

Figures are replaced to have complete data required

L 197 – homogeneity of data should be discussed

L 204 –all effects? What effects?

Comment

L237-238- to method part as well

Response

Yes, revised as recommended.

Comment

L 258 – and what is the conclusion from this part?

Response

Yes, revised as recommended.

Comment

L 278 – these effects were not discussed. Limitation part of the discussion is lacking.

Response

Yes, limitations were added as recommended.

Comment

Conclusions are weak.

Response

Conclusion was revised as recommended.